# Different Impacts of Heat-Killed and Viable *Lactiplantibacillus plantarum* TWK10 on Exercise Performance, Fatigue, Body Composition, and Gut Microbiota in Humans

**DOI:** 10.3390/microorganisms10112181

**Published:** 2022-11-03

**Authors:** Chia-Chia Lee, Yi-Chu Liao, Mon-Chien Lee, Yi-Chen Cheng, Shiou-Yun Chiou, Jin-Seng Lin, Chi-Chang Huang, Koichi Watanabe

**Affiliations:** 1Culture Collection & Research Institute, SYNBIO TECH INC., Kaohsiung 82151, Taiwan; 2Graduate Institute of Sports Science, National Taiwan Sport University, Taoyuan 333325, Taiwan; 3Department of Animal Science and Technology, National Taiwan University, Taipei 10672, Taiwan

**Keywords:** probiotics, postbiotics, viable, heat-killed, *Lactiplantibacillus plantarum* TWK10, exercise performance, anti-fatigue, microbiota

## Abstract

*Lactiplantibacillus plantarum* TWK10, a probiotic strain, has been demonstrated to improve exercise performance, regulate body composition, and ameliorate age-related declines. Here, we performed a comparative analysis of viable and heat-killed TWK10 in the regulation of exercise performance, body composition, and gut microbiota in humans. Healthy adults (*n* = 53) were randomly divided into three groups: Control, TWK10 (viable TWK10, 3 × 10^11^ colony forming units/day), and TWK10-hk (heat-killed TWK10, 3 × 10^11^ cells/day) groups. After six-week administration, both the TWK10 and TWK10-hk groups had significantly improved exercise performance and fatigue-associated features and reduced exercise-induced inflammation, compared with controls. Viable TWK10 significantly promoted improved body composition, by increasing muscle mass proportion and reducing fat mass. Gut microbiota analysis demonstrated significantly increasing trends in the relative abundances of *Akkermansiaceae* and *Prevotellaceae* in subjects receiving viable TWK10. Predictive metagenomic profiling revealed that heat-killed TWK10 administration significantly enhanced the signaling pathways involved in amino acid metabolisms, while glutathione metabolism, and ubiquinone and other terpenoid-quinone biosynthesis pathways were enriched by viable TWK10. In conclusion, viable and heat-killed TWK10 had similar effects in improving exercise performance and attenuating exercise-induced inflammatory responses as probiotics and postbiotics, respectively. Viable TWK10 was also highly effective in regulating body composition. The differences in efficacy between viable and heat-killed TWK10 may be due to differential impacts in shaping gut microbiota.

## 1. Introduction

Probiotics are defined as “live microorganisms that, when administered in adequate amounts, confer a health benefit on the host” [1]. There is accumulating evidence supporting the health benefits of probiotic consumption, including improvement of gastrointestinal function, modulation of immune and mental functions, and reduction in the risks of genital infection, cardiovascular disease, and other metabolic disorders [2,3,4]. Generally, probiotics improve the health status of hosts by enhancing intestinal barrier integrity, regulating the immune system, improving gut microbial composition, competitive exclusion of pathogens, reducing the intestinal pH, and increasing short-chain fatty acids (SCFAs), mucus, and bacteriocin production [5,6].

Emerging evidence supports the role of probiotics supplementation in improving the health status and exercise performance of elite athletes and the general population [7,8,9,10,11,12]. Regular exercise has many health benefits, including a reduced risk of cardiovascular disease, type 2 diabetes, and cancer, as well as lowering the likelihood of early death; however, contracting skeletal muscles generate free radicals, and prolonged and intense exercise can trigger oxidative stress accumulation and cause oxidative muscle tissue damage [13]. Exercise functions as a stressor and can cause inflammation. Acute exercise initiates a complex cascade of inflammatory events, which depend on exercise type, intensity, and duration. Pro-inflammatory cytokines, such as TNF-α, IL-1β, and IL-6, are released after sufficient physical activity. Several probiotics are reported to reduce exercise-associated oxidative stress and inflammatory responses [14,15,16]. 

Probiotics exert health benefits through the modulation of gut microbiota [17,18]. The relationship between the composition and metabolic activity of the gut microbiota, and exercise performance has gradually become a focus of attention [19,20]. In the colon, dietary fibers are digested by intestinal microbes and fermented into SCFAs, such as acetate, propionate, and butyrate, which are physiological energy sources and important modulators of metabolism, gut permeability, inflammatory responses, immune function, and exercise performance [21,22,23,24].

Live probiotic bacteria are affected by various host-specific factors in the gastrointestinal tract, and viability control is an issue requiring consideration when live probiotics are applied in the food and pharmaceutical industries. Inactivated probiotics have advantages relative to live probiotics; they do not translocate from the gut lumen to the blood or acquire and transfer antimicrobial resistance genes and are easy to transport and store for long periods [25,26,27]. Therefore, numerous studies have been conducted using inactivated microorganisms designated, “heat-killed probiotics”, “paraprobiotics”, “non-viable probiotics”, and “tyndallized probiotics”, with the aim of assessing their health benefits [27,28,29,30,31]. Inactivated (heat-killed) probiotic bacteria also have effects in modulating gut microbiota [30,32,33]. Comparative studies have demonstrated that non-viable probiotic products show similar potential health benefits to viable bacteria [34,35,36]. Accordingly, the focus in probiotic supplementation is gradually shifting from viable bacteria towards non-viable bacteria [37]. Recently, the International Scientific Association of Probiotics and Prebiotics has proposed the term “postbiotics” which encompasses inactivated microorganisms and stated its definition as “preparation of inanimate microorganisms and/or their components that confers a health benefit on the host” [38].

We have previously demonstrated that the administration of viable *Lactiplantibacillus plantarum* TWK10 (TWK10) exerted health benefits as a probiotic by improving exercise performance, increasing muscle mass and strength, changing body composition towards a healthy configuration, and ameliorating age-associated cognitive decline and impairment in mice and humans [11,12,39,40,41]. Nevertheless, the impacts of heat-killed TWK10 on health promotion are unknown. In this study, we investigated the effects of viable and heat-killed TWK10 in improving exercise performance, reducing fatigue, and modulating body composition and gut microbiota in humans.

## 2. Materials and Methods

### 2.1. Preparation of L. plantarum TWK10

TWK10 was isolated from a traditional Taiwanese pickled cabbage, “Po-tsai” as *Lactobacillus plantarum* [39], and identified as *Lactiplantibacillus plantarum* subsp. *plantarum*, by whole-genome sequencing [42,43]. TWK10 was cultivated and produced by SYNBIO TECH INC. (Kaohsiung, Taiwan) in capsule format with indicated doses. Heat-killed TWK10 cells were prepared by heating liquid bacterial culture at 70 °C for 60 min, and bacterial cell pellets were collected by centrifugation and spray-dried for capsule preparation. Each capsule contained either 1 × 10^11^ colony-forming units (CFU) of lyophilized TWK10 or 1 × 10^11^ heat-killed TWK10 cells (corresponding to 1 × 10^11^ CFU of TWK10) and was standardized with maltodextrin and microcrystalline cellulose. The ingredients of the placebo capsule were the same as those in the TWK10 capsule but without the addition of TWK10. The presence of no viable bacteria in the heat-killed TWK10 and placebo capsules was confirmed by cultivation, using de Man, Rogosa and Sharp (MRS; BD Difco, Franklin Lakes, NJ, USA) agar plates incubated anaerobically at 37 °C for 48 h.

### 2.2. Subjects

A total of 53 healthy subjects (26 men and 27 women; age, 20–30 years old) without professional athletic training were recruited in this study. Subjects were excluded from this study if they had smoking or drinking habits; were pregnant or planning pregnancy; or had any known disorders, including heart/cardiopulmonary disease, diabetes, neuromuscular disorder, neurological disease, autoimmune disease, peptic ulcers, ulcerative colitis, or other chronic diseases. All subjects were requested to maintain their usual diet and lifestyle, and were prohibited from consuming any other nutritional supplements, including probiotics, prebiotics, fermented products (yogurt or other foods), vitamins, minerals, herbal extracts, or antibiotics, to avoid interference during supplementation. Subjects who agreed to follow the study protocol and voluntarily signed informed consent were included in this study. The study was reviewed and approved by the Institutional Review Board of Landseed International Hospital (Taoyuan, Taiwan; LSHIRB No. 18-004-A2). The basic demographic characteristics of the subjects are presented in Table 1.

### 2.3. Experimental Design

This was a double-blinded and placebo-controlled trial, with a 2-week wash-out period and a 6-week intervention period. Eligible subjects were equally assigned to three groups (8–9 male and 9 female subjects in each group), including Control (placebo, 3 capsules/day), TWK10 (viable TWK10, 3 × 10^11^ CFU/day), and TWK10-hk (heat-killed TWK10, 3 × 10^11^ cells/day), based on individual exercise capacity, determined from the basal value of maximal oxygen consumption (VO_2max_). Maximal oxygen consumption and exercise performance were evaluated using a treadmill (Pulsar, h/p/cosmos, Nussdorf-Traunstein, Germany) and an auto respiratory analyzer Vmax 29c (Sensor Medics, Yorba Linda, CA, USA). Running speed on the treadmill started at 7.2 km/h and increased by 1.8 km/h every 2 min until volitional fatigue, according to the Bruce protocol [44]. Oxygen consumption was considered maximum when the respiratory exchange ratio (volume ratio of carbon dioxide produced to oxygen consumed; VCO_2_/VO_2_) was >1.10 and the maximum heart rate was achieved (maximum heart rate = 220 − age). VO_2max_ was used as a reference to adjust individual appropriate exercise intensity for physiological adaptation (60% VO_2max_) and exhaustive endurance performance (85% VO_2max_) tests. Adjustment of exercise intensity was calculated according to a previously described formula [11]. Subjects were required to avoid any strenuous physical activity for three days before VO_2max_ assessment and exercise tests. Endurance performance was assessed with a warm-up stage for 5 min, followed by an exercise test on the treadmill at 85% VO_2max_ workload. Oxygen consumption, heart rate, and Borg’s rating of perceived exertion scale were monitored every 5 min during submaximal endurance exercise, to determine achievement of exhaustion. Sustained exercise duration was recorded as the endurance index.

During the 6-week experimental period, subjects were required to take one capsule three times daily after meals, and to maintain their regular lifestyles. Information of caloric intake before and after administration were recorded as reference values. Physiological adaptation effects were determined before and after administration. Fresh fecal samples were collected for gut microbiota analysis and SCFA measurement on the last day of the 2-week wash-out period and after 6 weeks of administration.

### 2.4. Fatigue-Associated Biochemical Indices and Hematology Profiling

For assessment of fatigue-related indices, blood samples were collected at the indicated time points, just before and immediately after the 6-week experimental period, including baseline (0), 5 min (E5), 10 min (E10), 15 min (E15), and 30 min (E30) during the 60% VO_2max_ fixed intensity exercise challenge, and at 20 min (R20), 40 min (R40), 60 min (R60), and 90 min (R90) after exercise challenge. All biochemical indices were assessed using a Hitachi 7060 automatic biochemical analyzer (Hitachi, Tokyo, Japan). Complete blood count (CBC) profiles were determined at 90 min into the recovery phase (R90), using an automatic analyzer (MindrayBC-2800Vet, Shenzhen, China).

### 2.5. Body Composition

Body composition was measured by applying the multi-frequency principle with a bioelectrical impedance analyzer (BIA) on the InBody 770 (In-body, Seoul, Korea). This device takes 30 impedance measurements with frequencies of 1, 5, 50, 260, 500, and 1000 kHz for approximately 60 s. Before testing, age, sex, and height were entered for each subject. Subjects cleaned their hands and feet before contacting the electrodes and were then requested to stand on the center of electrodes and grasp the hand electrodes with their arms held so that there was no contact between the arms and the torso. The position was held for the duration of the test. Subjects fasted for at least 8 h prior to the tests.

### 2.6. DNA Extraction

Freshly collected fecal samples were washed three times with phosphate-buffered saline and centrifuged at 14,000× *g* for 5 min for extraction of bacterial genomic DNA. Fecal pellets were resuspended in 180 μL TE buffer containing lysozyme (final conc. 10 mg/mL), a suspension of glass beads (300 mg, 0.1 mm in diameter; Biospec, Bartesville, OK, USA) was added, and samples homogenized for 30 s using a FastPrep 24 homogenizer (MP Biomedicals, USA), to ensure complete disruption of cell walls and release of DNA molecules into the solution. Bacterial genomic DNA was then extracted using a Genomic DNA Mini Kit (Geneaid, Taipei, Taiwan), according to the manufacturer’s instructions. DNA concentrations were determined by spectrophotometry using a BioDrop instrument (Biochrom, Biochrom Ltd., Cambridge, UK). DNA samples were stored at −20 °C until further processing.

### 2.7. 16S rRNA Gene Sequencing and Analysis

The V3–V4 region of the 16S rRNA gene was amplified using specific primers (319F: 5ʹ-CCTACGGGNGGCWGCAG-3ʹ and 806R: 5ʹ-GACTACHVGGGTATCTAATCC-3ʹ) [45], according to the 16S Metagenomic Sequencing Library Preparation procedure (Illumina). Amplicon pools were sequenced on the Illumina MiSeq™ sequencing platform (Illumina, San Diego, CA, USA). Raw FASTQ files were initially demultiplexed using the q2-demux plugin and minimally quality filtered with DADA2 [46] (via q2-dada2), using Qiime2-2020.08 [47], to generate amplicon sequence variants (ASVs). Taxonomy of ASVs was performed using the q2-feature-classifier [48] and the classify-sklearn naïve Bayes taxonomy classifier, with the SILVA database (release 138), to identify representative sequences with 99% similarity [49]. Both alpha diversity (Shannon and Richness indices) and beta diversity were estimated using QIIME2, with a rarefaction of 30,000 sequences. Beta diversity analysis was performed using non-metric multidimensional scaling (NMDS) plots, based on weighted UniFrac or unweighted UniFrac distances. Permutational multivariate analysis of variance (PERMANOVA)/Adonis tests were conducted using vegan: Community Ecology Package (R package version 2.5-6; http://CRAN.R-project.org/package=vegan, accessed on 15 November 2021). Based on the characteristics of the compositional data, networks of specific families in each group were built using SparCC correlation coefficients [50]. Networks were visualized using Cytoscape (version 3.8.2; https://github.com/cytoscape/cytoscape/releases/3.8.2/, accessed on 23 November 2021). The Kyoto Encyclopedia of Genes and Genomes (KEGG; https://www.genome.jp/kegg/, accessed on 6 December 2021) database was used to analyze pathway enrichment, using Phylogenetic Investigation of Communities by Reconstruction of Unobserved States (PICRUSt2) [51]. Finally, the influence of each differentially abundant gut microbial component was evaluated by linear discriminant analysis (LDA) to determine effect size (LEfSe) [52]. Raw sequence files supporting the findings of this article are deposited in the NCBI Sequence Read Archive (SRA) database, with project accession number, PRJNA791018.

### 2.8. SCFA Levels in Feces

Freshly collected fecal samples were mixed with 70% ethanol solution at a ratio of 1 mg of fecal sample: 10 μL 70% ethanol, and then homogenized with appropriate amounts of glass beads (1.0 mm in diameter; Biospec Products) by vortexing at 3000 rpm for 10 min. Homogenized samples were centrifuged at 14,000× *g* for 10 min, and the supernatants were collected for fatty acid derivatization, according to a previously described method [53]. Derivatized supernatants were filtered using a 0.22-μm polycarbonate syringe filter (Millipore, St. Charles, MO, USA). SCFAs were separated and quantified using high-performance liquid chromatography (HITACHI, Tokyo, Japan) on a C18 HTec column (NUCLEODUR, Macherey-Nagel, Düren, Germany), with column temperature 40 °C, flow rate 1 mL/min, and detection wavelength 400 nm.

### 2.9. Statistical Analysis

Data are expressed as mean ± SD. Statistical analysis was performed using GraphPad Prism 8.1.1 (GraphPad Software, San Diego, CA). Statistical differences among groups were analyzed by two-way repeated-measures ANOVA with Tukey *post-hoc* test. Differences between before and after administration were analyzed by two-way repeated-measures ANOVA with Bonferroni *post-hoc* test. Differences in the changes between before and after administration in each subject among the three groups were analyzed by Kruskal–Wallis test with Dunn *post-hoc* test. Spearman’s correlation coefficient was used for analyses of correlations between gut microbial abundances and exercise-associated phenotypic features. *P* < 0.05 was considered statistically significant.

## 3. Results

### 3.1. Both Viable and Heat-Killed TWK10 Improved Exercise Endurance Performance and Physical Adaptation

In this study, we assessed the effects of viable and heat-killed TWK10 on exercise endurance performance, which was evaluated by a time-to-exhaustion test with an 85% VO_2max_ workload. There were no significant differences in the basic demographic characteristics or baseline exhaustion time values of subjects among the three groups (Control, TWK10, and TWK-hk) (Table 1). At week 0, there were no significant differences in mean exhaustion time among the three groups; however, after six weeks of administration, the mean exhaustion times in the TWK10 (17.55 ± 3.98 min; *P* < 0.001) and TWK10-hk (16.72 ± 5.91 min; *P* < 0.01) groups were significantly higher than those in the Control group (12.23 ± 2.08 min). Within-group comparisons demonstrated that, after 6 weeks of treatment, mean exhaustion times in the TWK10 and TWK10-hk groups were significantly increased by 1.38-fold (*P* < 0.001) and 1.33-fold (*P* < 0.001), respectively, whereas no significant difference was observed in the Control group. These results show that both viable and heat-killed TWK10 improved exercise endurance performance, while there was no significant difference in the level of improvement between the TWK10 and TWK10-hk groups (Figure 1A).

Next, we assessed fatigue-related indices and found that there were no significant differences in concentrations of lactate, ammonia, glucose, and creatine kinase in plasma from all three groups during and after exercise challenges just before administration (data not shown), whereas the concentrations of lactate and ammonia in plasma from all the three groups immediately after 6-week of administration were increased during exercise challenge, reaching peak levels after 30 min of exercise stimulation (time point: E30), and gradually decreased to the basal level during the recovery phase. In the TWK10 and TWK10-hk groups, lactate and ammonia levels in the exercise phase [for lactose (time points: E10, E15, and E30); for ammonia (time points: E15 and E30)] and in the recovery phase [for lactate (time points: R20 and R40); for ammonia (time point: R20)] were significantly lower than those in the Control group (Figure 1B,C). Lactate concentrations in the TWK10 group in the exercise phase at time points E15 and E30 were significantly lower (*P* < 0.05) than those in the TWK10-hk group (Figure 1B). Further, plasma glucose concentrations in the TWK10 group at time points E15, E30, and R20 were significantly higher (*P* < 0.05) than those in both the TWK10-hk and Control groups. After exercise stimulation, the elevated glucose level was gradually reduced and returned to basal level at resting state (Figure 1D). For creatine kinase (CK), no significant differences among the three groups were observed during the exercise test (Figure 1E).

### 3.2. Heat-Kill TWK10 Was More Effective on Reducing Exercise-Induced Inflammatory Response

The neutrophil to lymphocyte ratio (NLR) and platelet to lymphocyte ratio (PLR) serve as markers of systemic inflammatory response in humans [54,55]. Therefore, to evaluate the effect of TWK10 on the exercise-induced systemic inflammatory response, NLR and PLR were examined. CBC was determined from blood samples collected at time point R90 (90 min in recovery state after 30-min exercise stimulation at 60% VO_2max_) and NLR and PLR were calculated. Before administration, no significant differences in PLR and NLR values were observed among the three groups. Following administration of heat-killed TWK10 for 6 weeks, NLR and PLR in the TWK10-hk group were significantly decreased (*P* < 0.05 and *P* < 0.01, respectively), whereas, in subjects who received viable TWK10 for 6 weeks, no significant alteration in NLR was observed; however, a significant reduction in PLR l (*P* < 0.05) was detected (Figure 2).

### 3.3. Viable TWK10 Was More Effective on the Modulation of Muscle Weight and Body Fat Mass

To assess the impact of viable and heat-killed TWK10 on modulation of body composition, muscle weight, body fat mass, and body mass index (BMI) of subjects before and after administration were measured using an InBody 770 Body Composition Analyzer. Considering individual within-group differences, we calculated the changes after six weeks of treatment, compared with before treatment, for each subject, then compared these changes among the three groups. After six weeks of administration, there were no significant differences in mean muscle weight among the Control, TWK10, and TWK10-hk groups; however, a significant increase (*p* < 0.001) in muscle weight was observed in the TWK10 group following treatment. Next, we compared differences in muscle weight before and after treatment in each subject among the three groups. A significant increase of mean muscle weight was observed in the TWK10 group after six weeks of treatment, relative to that in the Control group (*p* < 0.001). Further, the mean fat mass (%) of subjects in the TWK10 group after administration was significantly lower (*p* < 0.001) than that before administration, and a significant difference (*p* < 0.001) in the decrease in mean fat mass after six weeks of administration was observed in the TWK10 group relative to the Control group. No significant difference in the change in BMI was observed among the three groups (Table 2).

### 3.4. Viable and Heat-Killed TWK10 May Trigger Distinct Gut Microbial Community Changes

To further understand the impact of TWK10 on gut microbial composition, we analyzed the characteristics of bacterial community composition by high-throughput sequencing of the V3–V4 region of the 16S ribosomal RNA gene. To determine how the overall profile of microbial composition was modulated by administration of viable or heat-killed TWK10, alpha- and beta-diversity indices were analyzed. Median Shannon index values for α-diversity in each group ranged from 3.2 to 3.5, and no significant differences were observed between values before and after administration in each group, nor among the three groups. 

The median observed ASVs ranged from 130 to 170, and no significant differences were observed between before and after administration in each group, nor among the three groups (Figure 3A). To evaluate differences in microbial community structures in each group before and after administration among the three groups, we generated NMDS ordination plots of unweighted and weighted Unifrac distances for all six experimental groups. There were no significant differences between before and after administration, nor among the three groups. After six weeks of administration, the β-diversity profiles of fecal microbial composition in subjects showed a strong trend toward significant difference between the TWK10 and TWK10-hk groups (PERMANOVA, *P* = 0.063), based on NMDS analysis with unweighted UniFrac metric (Figure 3B). Box plots of unweighted UniFrac distances showed that gut communities in the TWK10 group after administration were significantly different from those of the TWK10-hk group after administration (*p* < 0.0001) (Figure 3C). β-diversity profiles in the TWK10 group after administration showed a marginally significant difference (*p* = 0.072, by unweighted UniFrac distance) to those in the Control group after administration (Figure 3C), while those in the TWK10-hk group differed significantly from those in the Control group (PERMANOVA, *P* = 0.036, by NMDS with unweighted UniFrac metric; *P* = 0.072 by unweighted UniFrac distance) (Figure 3B,C). β-diversity profiles in the TWK10-hk group differed significantly different from those before administration (*p* < 0.0001, by weighted UniFrac distance; *P* = 0.031, by unweighted UniFrac distance), whereas those in the TWK10 group did not differ significantly before and after administration (Figure 3C).

The overall microbiota structures of subjects in the three groups before and after administration at the phylum level are presented in Figure 3D. The predominant phyla detected in subjects were *Firmicutes*, *Bacteroidota*, *Actinobacteriota*, *Proteobacteria*, and *Verrucomicrobiota*. After six weeks of administration, a weak increasing trend in the relative abundance of *Verrucomicrobiota* in the TWK10 group (*P* = 0.114), and significant increases in the relative abundances of *Proteobacteria* in the Control (*P* = 0.011) and TWK10-hk (*P* = 0.030) groups were observed, relative to those before administration (Appendix A). At the family level, the top 10 predominant bacteria in fecal samples from subjects were *Lachnospiraceae*, *Bacteroidaceae*, *Ruminococcaceae*, *Bifidobacteriaceae*, *Prevotellaceae*, *Selenomonadaceae*, *Coriobacteriaceae*, *Veillonellaceae*, *Acidaminococcaceae*, and *Streptococcaceae* (Figure 3E). After 6 weeks of administration in the TWK10 group, significant decreasing trends in the abundance of *Ruminococcaceae* (*P* = 0.109) and *Eggerthellaceae* (*P* = 0.053), a significant increase in the abundance of *Lactobacillaceae* (*P* = 0.039), and significant increasing trends in the relative abundances of *Prevotellaceae* (*P* = 0.109) and *Akkermansiaceae* (*P* = 0.114) were observed, relative to those before administration. In the TWK10-hk group after 6 weeks of administration, a significant increasing trend (*P* = 0.072) in the abundance of *Enterobacteriaceae*, a significant decrease (*P* = 0.030) in the abundance of *Peptostreptococcaceae*, a significant decreasing trend (*P* = 0.107) in the abundance of *Lachnospireaceae*, and an increasing trend (*P* = 0.148) in the abundance of *Atopobiaceae* were observed as compared with before administration (Appendix A). In the TWK10 group after six weeks of administration, genus-level analysis revealed a significant decrease in *Lachnospira* abundance (*P* = 0.022), a decreasing trend in *Faecalibacterium* abundance (*P* = 0.077), and a significant increasing trend in *Akkermansia* abundance (*P* = 0.114), compared with before administration. Meanwhile, after six weeks of administration in the TWK10-hk group, we detected significant increases in the abundances of *Lactococcus* (*P* = 0.046) and *Escherichia-Shigella* (*P* = 0.038), decreasing trends in the abundances of *Roseburia* (*P* = 0.110) and *Lachnospira* (*P* = 0.064), and a marginal increase in the abundance of *Oscillibacter* (*P* = 0.184), relative to before administration (Appendix A).

### 3.5. Viable and Heat-Killed TWK10 Showed Different Imapcts on Gut Microbial Co-Occurrence Networks

As intra-microbiota bacterial interactions play roles in shaping the gut microbiota community [56], we further investigated alterations in bacteria–bacteria interactions after viable or heat-killed TWK10 administration using bacterial co-occurrence networks analysis. Among the ASVs obtained in the study, 16S rRNA read counts were >50 in 233 and 258 ASVs before administration in the TWK10 and TWK10-hk groups, respectively. SparCC correlation coefficients >|0.6| and *P* < 0.05 were considered indicative of a connection between bacterial groups. Gut bacteria co-occurred more frequently before administration than after administration in both the TWK10 and TWK10-hk groups. Specifically, in the TWK10 and TWK10-hk groups, 37 and 30 nodes, respectively, were observed before administration, whereas 28 and 25 respective nodes were observed after administration. These decreases reflected dramatic changes in bacterial network structures in response to the administration of viable TWK10 (Figure 4A,B). In the TWK10 group before administration, *Lachnospiraceae* was positively correlated with *Butyricicoccaceae*, *Carnobacteriaceae*, and *Sutterellaceae*, whereas it was negatively correlated with *Anaerovoracaceae* and *Christensenellaceae*; *Ruminococcaceae* was positively correlated with *Bacteroidaceae*, *Butyricicoccaceae*, *Carnobacteriaceae*, and *Desulfovibrionaceae*; and *Bifidobacteriaceae* was positively correlated with *Veillonellaceae* (Figure 4A). Meanwhile, in the TWK10 group after administration, *Lachnospiraceae* was positively correlated with *Actinomycetaceae*, *Butyricicoccaceae*, *Carnobacteriaceae*, and *Streptococcaceae*; and *Bifidobacteriaceae* was positive correlated with *Bacteroidaceae* and *Enterobacteriaceae*, and negatively correlated with *Eubacterium coprostanoligenes* group, *Marinifilaceae*, and *Oscillospiraceae*. In addition, a firm network among the families, *Anaerovoracaceae*, *Christensenellaceae*, *Coriobacteriales incertae sedis*, *E. coprostanoligenes* group, *Marinifilaceae*, *Oscillospiraceae*, and *Rikenellaceae* was observed (Figure 4B). In the TWK10-hk group before administration, *Lachnospiraceae* was positively correlated with *Butyricicoccaceae*, whereas it was negatively correlated with *Christensenellaceae*, *Coriobacteriales incertae sedis*, and *Oxalobacteraceae*; a positively correlated network among *Carnobacteriaceae*, *Eggerthellaceae*, and *Streptococcaceae* was observed; and *Prevotellaceae* was negatively correlated with *Bacteroidaceae*, *Bifidobacteriaceae*, and *Monoglobaceae* (Figure 4C). However, heat-killed TWK10 administration altered the topology of the co-occurrence network, with the number of connections among gut bacteria reduced. Specifically, *Lachnospiraceae* was positively correlated with *Eggerthellaceae* and *Sutterellaceae*, and negatively correlated with *Christensenellaceae*; *Ruminococcaceae* was positively correlated with *Bifidobacteriaceae* and *Butyricicoccaceae*, and negatively correlated with *Erysipelotrichaceae*; and *Anaerovoracaceae*, *Desulfovibrionaceae*, *Marinifilaceae*, and *Rikenellaceae* formed a positively correlated network (Figure 4D).

### 3.6. Viable and Heat-Killed TWK10 Showed Differenet Impacts on Predicted Gut Microbial Community Functional Profiles

Functional profiles of bacterial communities of the three groups (Control, TWK10, and TWK10-hk) were predicted using PICRUSt2. LEfSe analysis was then performed to explore level 3 KEGG pathways with significant differences in abundance before and after administration. Butanoate metabolism, glutathione metabolism, and ubiquinone and other terpenoid-quinone biosynthesis pathways were over-represented (α = 0.1, LDA score > 2.0) following TWK10 administration (Figure 5A). In the TWK10-hk group, RNA polymerase, pentose phosphate pathway, sulfur relay system, galactose metabolism, beta_lactam resistance, and starch and sucrose metabolism were over-represented in the microbiota of subjects before administration, whereas phenylalanine metabolism, tryptophan metabolism, butanoate metabolism, and taurine and hypo-taurine metabolism were significantly enriched (α = 0.1, LDA score > 2.0) following administration (Figure 5B). These results demonstrate that viable and heat-killed TWK10 induced different alterations in the overall predicted functional features of gut microbiota.

### 3.7. Both Viable and Heat-Killed TWK10 Increased Gut SCFA Levels

Acetate, propionate, and butyrate are the main SCFAs metabolized by gut microorganisms. Therefore, we collected fecal samples from subjects before and after administration to examine whether TWK10 could influence SCFA production. As shown in Table 3, subjects receiving both viable and heat-killed TWK10 had significantly higher acetate concentrations in feces relative to before administration (*P* < 0.05). Further, a significantly increasing trend in propionate was detected in feces of subjects treated with heat-killed TWK10 (*P* = 0.0857), and in butyrate in feces of subjects treated with viable TWK10 (*P* = 0.0744). In the Control group, no significant alterations of any of the three tested SCFAs were observed after administration.

### 3.8. Correlation between Gut Microbial Composition and TWK10-Mediated Health Benefits

Correlations between the relative abundances of gut bacterial families and TWK10-mediated phenotypic features related to host health benefits were assessed by Spearman’s correlation analysis. For exercise endurance performance, the effects on exhaustion time mediated by administration of heat-killed TWK10 were significantly and positively correlated with the *Veillonellaceae* population, and significantly negatively correlated with *E. coprostanoligenes* group, *Erysipelatoclostridiaceae*, and *Lachnospiraceae* populations. In subjects receiving viable TWK10, exhaustion time was significantly positively correlated with the *Coriobacteriaceae* population. The correlation coefficients between *Veillonellaceae* and the exhaustion time in subjects administered viable and heat-killed TWK10 were strengthened from −0.35 to 0.06, and from 0.51 to 0.65, respectively, following administration. Changes in body fat mass (%) mediated by heat-killed TWK10 administration were significantly positively correlated with the presence of *Erysipelatoclostridiaceae* and *Erysipelotrichaceae*, and significantly negatively correlated with the *Veillonellaceae* population. In subjects administered viable TWK10, the abundances of *Bacteroidaceae*, *Oscillospiraceae*, *Rikenellaceae*, and *Ruminococcaceae* were negatively correlated with muscle weight. In pro-inflammatory responses detected following the administration of heat-killed TWK10, a strong positive correlation was observed between the abundance of *Enterobacteriaceae* and PLR, whereas *Akkermansiaceae* abundance was strongly negatively correlated with NLR (Figure 6).

## 4. Discussion

Viable TWK10 has been previously elucidated to have potential probiotic effects in enhancing exercise performance, increasing muscle mass and strength, improving body conformation, and ameliorating age-associated cognitive decline and impairments in mice and humans [11,12,39,40,41]; however, the impacts of heat-killed TWK10 on health promotion have yet to be confirmed. Therefore, in the current study, we applied the treadmill method to evaluate exercise performance before and after administration of viable or heat-killed TWK10 by assessing changes in individual VO_2max_. Six weeks of administration of both viable and heat-killed TWK10 significantly improved exercise performance, consistent with our previous findings in humans, where administration of viable TWK10 significantly increased exercise performance relative to the placebo group [11,12].

Blood glucose and glycogen stored in muscle are established as the major energy sources during exercise. Exercise can spike blood glucose, most commonly via the release of the stress hormone, adrenaline. Key actions of adrenaline include increasing heart rate and blood pressure, expanding the lung air passages, enlarging the pupils in the eye, redistributing blood to the muscles, and altering metabolism in response to acute stress [57]. Generally, exercise induces an increase in gluconeogenesis and raises blood glucose, to provide fuel for muscles in response to exercise demand. After exercising, the body adjusts the blood glucose by secreting insulin, to lower the excess sugar in circulation that is no longer needed by the muscles. In this study, we observed a significant elevation in circulating glucose during exercise in subjects who received viable TWK10, reflected in an improvement in exercise endurance performance (Figure 1D). These results are consistent with our previous findings [11,12]. In contrast, no significant elevation of plasma glucose was observed in subjects who received the heat-killed form of TWK10; therefore, we speculate that viable TWK10 can accelerate cross-talk between the gut and brain, to efficiently trigger adrenaline release, boost energy supplies, and consequently prolong exercise duration. However, further investigations are needed to elucidate the role of TWK10 in adrenaline signaling and improved exercise performance.

In addition to the exercise-induced boost in blood glucose, significantly reduced lactate production in circulation was also observed in subjects who received viable TWK10 (Figure 1B). Muscle contraction depends on the breakdown of adenosine triphosphate (ATP) and the concomitant release of free energy. Anaerobic glycolysis is the major metabolic pathway used when oxygen supply is limited during exercise, such as high-intensity, sustained, isometric muscle activity. Glycolysis produces pyruvate from glucose, which is then reduced to lactate-by-lactate dehydrogenase, without oxygen consumption [58]. In groups administered both viable and heat-killed TWK10, exercise endurance (time to exhaustion) was improved and less plasma lactate was produced during exercise, indicating that subjects receiving TWK10 were less likely to obtain energy for exercise demand through anaerobic glycolysis. Sufficient oxygen supply in exercising muscle promotes aerobic glucose breakdown, resulting in the conversion of pyruvate to acetyl-CoA, which is subsequently metabolized in the TCA cycle to produce ATP for exercise demand. Enhanced TCA cycle activity results in reduced lactate production. In KEGG analysis, we detected an increasing trend in TCA cycle enrichment (*P* = 0.069) in subjects receiving heat-killed TWK10, relative to the baseline value (data not shown). Another possible reason for the observation of reduced levels of lactate during exercise is an alteration of energy supply. Ketosis decreases muscle glycolysis and plasma lactate levels, while increasing intramuscular triacylglycerol oxidation during exercise, providing an alternative substrate for oxidative phosphorylation [59]. Further investigations are necessary to understand the effects of both viable and heat-killed TWK10 on energy metabolism during exercise.

There is good evidence that inactivated microbial cells exhibit health-promoting effects through immune system modulation, prevention of pathogenic infection, and reduction in oxidative stress [60,61,62]. Heat-killed *Lactobacillus brevis* SBC8803 downregulates the expression of pro-inflammatory cytokines and enhances intestinal barrier function under oxidative stress [63]. Likewise, heat-killed *Bacillus coagulans* GBI-30 promoted immune responses and modulated inflammatory cytokine expression in cell-base assays [64]. It is established that both viable or heat-killed bacteria can exhibit anti-inflammatory effects; however, which form of bacteria is more effective depends on the strain. In this study, subjects receiving both viable and heat-killed TWK10 showed significant reduction in PLR post-exercise challenge, whereas significantly reduced NLR was observed only in subjects who received heat-killed TWK10 (Figure 2), indicating that heat-killed TWK10 has the potential to reduce damage or increase recovery rate after exercise. Exercise-induced production of muscle damage indicators and inflammatory biomarkers can lead to a temporary reduction in muscular force [65] and decreased physical performance [66]. These findings suggested that the improvement of exercise performance mediated by viable and heat-killed TWK10 may be achieved in a different manner. Therefore, further investigations are needed to further elucidate the differences in underlying mechanisms.

Total daily energy expenditure is determined by basal metabolic rate (BMR), food-induced thermogenesis, and energy required for physical activity. Generally, the BMR accounts for 65–75% of total energy expenditure and is considered to be proportional to fat-free mass [67]. As Huang et al. previously reported [68], viable TWK10 upregulates BMR in gnotobiotic mice, accompanied by a reduction in fat mass, without altering dietary intake. In addition, the proteomic analysis demonstrated that peroxisomal acyl-coenzyme A oxidase 2 and very long-chain acyl-CoA synthetase were significantly upregulated in the liver of mice administered viable TWK10, indicating that lipid metabolism is enhanced by TWK10 [69]. In this study, viable TWK10 significantly promoted the development of body composition toward an improved configuration by increasing muscle mass proportion and reducing fat mass in humans (Table 2), consistent with our previous findings [12]. These results suggest that the increase in muscle mass mediated by viable TWK10 may contribute to the upregulation of energy expenditure while promoting lipid metabolism and further reducing body fat accumulation. In addition, there is evidence that gut microbiota are directly involved in regulating energy metabolism; thus, changes in the composition and abundance of gut bacteria may modify energy consumption and expenditure [70]. To better understand the mechanisms underlying the regulation of body composition by viable and heat-killed TWK10, further investigation of energy metabolism and gut microbiota is needed.

Although the relationships between diet, gut microbiota, host immunity, and host metabolism are becoming more evident [71,72,73], that between microbiota and exercise has not been fully investigated. Many studies have shown that the consumption of probiotics has the potential to positively modify gut microbiota community structures, which may be important to increase exercise performance in physical activity practitioners and athletes [74,75,76,77]; however, the potential mechanisms by which probiotic strains may modulate gut microbiota profiles to improve exercise performance remain unclear. Ecological diversity of microbiota is important for promoting health stability and exercise performance. Microbiota alpha-diversity has been linked to human health, with loss of diversity associated with several conditions, including autism, gastrointestinal diseases, and obesity-associated inflammatory characteristics [78]. In this study, no differences in alpha-diversity of the gut microbiota were observed before and after six weeks of administration in any of the three groups; however, the number of gut microbial taxa (mean observed ASVs) tended to be higher in the TWK10-hk group after administration than that in the Control group (Figure 3A), strongly suggesting that the significant difference (*P* = 0.036) in beta-diversity, based on NMDS with unweighted UniFrac metric, in the TWK10-hk group after administration, relative to that in the Control group, was due to an increase in ASVs in the TWK10-hk group. Together with detection of a highly significant (*P* < 0.0001, by unweighted UniFrac) difference in beta-diversity distance in the TWK10-hk group after administration relative to the TWK10 group, these findings indicate that six-week administration of heat-killed TWK10 has a stronger effect in rearranging microbial community structures than viable TWK10 (Figure 3B,C).

At the bacterial phylum level, no obvious alterations in predominant gut microbiota, except for *Verrucomicrobiota* and *Proteobacteria*, were detected following six-week administration of viable or heat-killed TWK10. Relative abundance of the phylum, *Verrucomicrobiota*, a mucin-degrading bacteria that resides in the mucus layer and represents 1%–4% of fecal microbiota in healthy humans [79], was weakly significantly (*P* = 0.114) increased by administration of viable TWK10. Further, the administration of heat-killed TWK10 led to a significant (*P* = 0.030) increase in the relative abundance of the phylum, *Proteobacteria*, which mainly comprised the family *Enterobacteriaceae*. Nevertheless, the gut microbiota in both the viable and heat-killed TWK10 administration groups were more complex and apparently divergent from the Control group at the bacterial family and genus levels, representing differences in microbial profiles and well-characterized structures. *Verrucomicrobiota* was mostly comprised of the single genus, *Akkermansia* (family *Akkermansiaceae*), which showed a weak significant (*P* = 0.114) increase in the TWK10 group after administration (Appendix A). *Akkermansia muciniphila* is a mucin-degrading bacteria and its abundance is inversely correlated with obesity and associated metabolic disorders [80,81]. The proportions of the genus, *Akkermansia*, in athletes are significantly higher in those with low BMI [76], which is generally considered a healthier metabolic profile [82]; however, although we detected a significant increase in the relative abundance of *Akkermansia*, no significant reduction in BMI was observed in the TWK10 group after administration. Further, we detected a weakly significant (*P* = 0.109) increase of *Prevotellaceae*, predominantly comprising the genus, *Prevotella*, following the administration of viable TWK10. *Prevotella* has been identified as more universal in populations with plant-rich diets, abundant in carbohydrates and fiber [83], and *Prevotella* and *Akkermansia* produce acetate via the Wood-Ljungdahl and acetyl-CoA pathways [84,85].

Based on gut microbial co-occurrence network analysis, we found that the number of bacteria–bacteria interactions decreased in response to the administration of viable and heat-killed TWK10 (Figure 4). Following the administration of viable TWK10, three butyrate-producing bacterial families, *Lachnospiraceae*, *Butyricicoccaceae*, and *Actinomycetaceae*, were strongly and positively correlated with one another. In addition, the families, *Christensenellaceae*, *Eubacterium coprostanoligenes*, and *Oscillospiraceae*, of the phylum *Firmicutes*, and *Coriobacteriales incertae sedis*, of the phylum *Actinobacteria*, which are regarded as potentially beneficial bacteria [86,87,88,89], showed a strong positive correlation with one another. Meanwhile, following administration of heat-killed TWK10, acetate- and butyrate-producing bacteria, such as *Bifidobacteriaceae* (acetate), *Butyricicoccaceae* (butyrate), and *Ruminococcaceae* (butyrate), showed strong positive correlations with one another. Further, two butyrate-producing and potentially probiotic families, *Lachnospiraceae* and *Eggerthellaceae*, were positively correlated with one another. These findings demonstrate differences in the influence on the co-occurrence of gut microbial components between viable and heat-killed TWK10; however, both viable and heat-killed TWK10 were able to rearrange gut microbial community structures. In addition, it has been confirmed that the administration of probiotics increased the production of SCFAs by modulation of the gut microbiota [41,90,91]. Alterations in gut microbiota can stimulate differential production of SCFAs (e.g., butyrate and acetate), which play important roles in the maintenance of gut and metabolic health [92,93].

Spearman’s correlation analysis revealed that exercise-associated host phenotypic features were positively or negatively correlated with specific microbiota, and that those correlations differed between viable and heat-killed TWK10; that is, exhaustion time was positively correlated with *Veillonellaceae* in subjects administered heat-killed TWK10, and with *Coriobacteriaceae* in subjects administered viable TWK10. Relative abundance of *Veillonellaceae*, which produces propionate from lactate [94], was increased in athletes after running a marathon, and oral administration of the *Veillonella* strain significantly increased exhaustion time in mice [77]. In the current study, trends toward strengthened correlation between *Veillonellaceae* and exhaustion time were also observed in subjects administered both viable and heat-killed TWK10, relative to basal state. Body fat mass was positively correlated with the butyrate-producing bacteria, *Erysipelatoclostridiaceae* and *Erysipelotrichaceae*, in subjects administered heat-killed TWK10. Further, the significant reduction in NLR values in subjects administered heat-killed TWK10 was negatively correlated with a trend toward an increased relative abundance of *Akkermansiaceae*, which is regarded as an intestinal mucin-degrader [95] and effective in reducing inflammation [96]. Accordingly, to clarify the differences in effects on exercise performance between viable and heat-killed TWK10, it will be necessary to comprehensively analyze more samples to determine the effects of each TWK10 state (viable and heat-killed) on correlations between exercise-associated host phenotypic features and microbial structures.

Various studies have investigated the effects of the gut microbiome on exercise performance [77,97,98]. Exercise performance was improved in mice adapted with individual bacterial taxa, relative to their germ-free counterparts, indicating that increased microbial diversity has a beneficial effect on exercise. Recent studies have also shown that gut microbiota may be critical for skeletal muscle metabolism and host function [21,99]. Additionally, natural reseeding of the gut microbiota or infusion of acetate reversed the loss of endurance capacity and muscle contractile function in antibiotic-treated mice [21]. Further, the probiotics, *Streptococcus thermophilus* FP4 and *Bifidobacterium breve* BR03, attenuated performance decrements and muscle tension in the days following muscle-damaging exercise [100]. Nevertheless, while these studies showed that probiotics can modulate gut microbiota and improve exercise capacity, their effects on performance remain unclear.

To reveal the putative mechanisms underlying the probiotic effects of TWK10 in improving exercise performance, functional profiling was performed using PICRUSt2, based on gut microbial taxa. LEfSe analysis, based on differential functional abundances identified through KEGG pathway mapping, revealed that several metabolic pathways were significantly upregulated in subjects administered with viable or heat-killed TWK10 (Figure 5). Butanoate metabolism was elevated in groups treated with both viable and heat-killed TWK10. Increased butanoate metabolism results in butyric acid formation, which is a major SCFA that reduces inflammation and promotes gut health [101,102,103,104]. The upregulation of pathways, such as glutathione metabolism, ubiquinone and other forms of terpenoid-quinone biosynthesis, suggests that viable TWK10 can increase antioxidant defense and detoxification reactions, which protect the body from oxidative stress and have benefits on exercise. Increased metabolism of the non-enzymatic antioxidants, glutathione and ubiquinone (coenzyme Q10), leads to increased resistance to exercise-induced oxidative challenges [105,106]. In addition, glutathione supplementation induces aerobic metabolism and improves an acidic environment in skeletal muscle, which in turn prevents exercise-induced fatigue [107]. Meanwhile, KEGG pathway analysis of the heat-killed TWK10-administered group indicated the elevation of different metabolic pathways, including phenylalanine metabolism, tryptophan metabolism, and taurine and hypo-taurine metabolism. Phenylalanine is an essential gluconeogenic amino acid as well as a gluconeogenic and ketogenic, which becomes trans-aminated into different intermediates of the gluconeogenic pathway [108]. Phenylalanine supplementation can increase plasma glucagon concentrations during exercise. Glucagon is a key hormone involved in fat catabolism during exercise, suggesting that increased phenylalanine metabolism can stimulate fat oxidation through glucagon secretion [109]. Tryptophan is an essential amino acid which is metabolized through the kynurenine pathway to generate a number of bioactive substances, thereby modulating health and disease states, ranging from intestinal conditions to inflammation and cancer progression [110,111]. Kynurenine and its metabolites can mediate the effects of exercise, mood, and neuronal excitability and, ultimately, communicate with microbiota. Taurine is an amino acid that can regulate the gut micro-ecology and has the potential to enhance gut-resistance to pathogenic bacteria [112]. The results obtained in our study demonstrate that administration of viable and heat-killed TWK10 can influence the gut microbiota, which has emerged as an important driving force in modulating metabolic activities, although the magnitude, features, and strength of viable and heat-killed TWK10 were inconsistent.

The gut microbiota regulates multiple functions related to host physical health, and mental health through the gut-brain axis which is the two-way communication pathway between the enteric and central nervous systems [113]. Through these pathways, produced SCFAs, bile acids, and tryptophan by the gut microbiota interact with enteroendocrine cells and activate the vagus nerve which serves a critical role in communication between the gut microbiota and the brain [114,115]. Metabolomics is an emerging technology that could simultaneously quantify multiple types of small molecules, such as amino acids, fatty acids, carbohydrates, or other products of cellular metabolic functions [116]. Metabolome response closely associates with physical activity and biological functions [117,118]. Therefore, to determine the metabolome associations with TWK10-mediated phenotypic features could provide greater insights into the understanding of the underlying the different mechanisms of viable and heat-killed TWK10.

## 5. Conclusions

In the present study, we present, for the first time, evidence of the effects of TWK10 from a clinical trial using both viable and heat-killed TWK10, demonstrating that both exert sufficient probiotic and postbiotic effects, respectively, in improving exercise performance and fatigue-associated features, and mitigate responses to exercise-induced inflammation. Further, we detected differences in the regulation of body composition and anti-inflammation responses between viable and heat-killed TWK10. These differences may be due to differential impacts in shaping gut microbiota. Further studies are needed to clarify the differences in the efficacy of the two states of TWK10 in promoting exercise performance in clinical trials with more subjects.

## Figures and Tables

**Figure 1 microorganisms-10-02181-f001:**
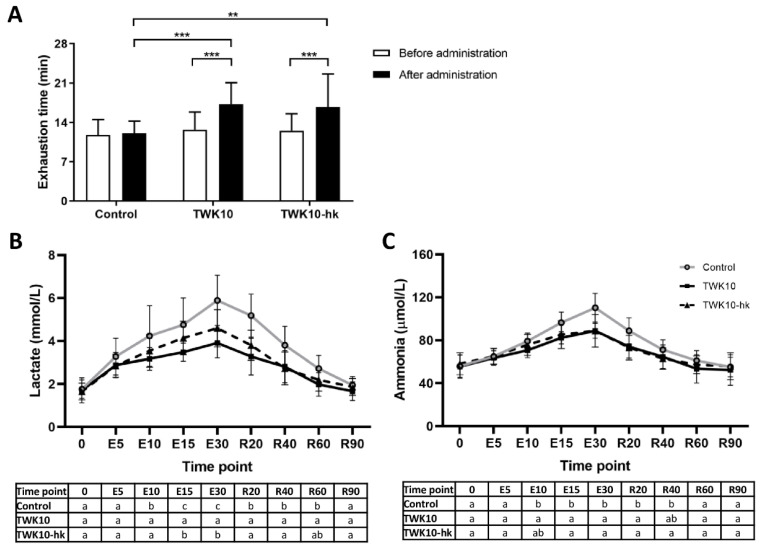
Effects of TWK10 on exercise endurance performance and fatigue-associated blood indicators. (**A**) Endurance performance was evaluated under 85% VO_2max_ exercise intensity before and after TWK10 administration. Statistical differences among groups were analyzed by two-way repeated-measures ANOVA with Tukey *post-hoc* test. The significance of differences between parameters before and after administration was analyzed by two-way repeated-measures ANOVA with Bonferroni *post-hoc* test. ** *P* < 0.01, *** *P* < 0.001. During fixed intensity and period exercise tests, blood samples were collected for (**B**) lactate, (**C**) ammonia, (**D**) glucose, and (**E**) CK measurements at the indicated time points after TWK10 administration. Data are presented as mean ± SD. Statistical differences among groups were analyzed by one-way ANOVA with Tukey *post-hoc* test. Different letters (a, b, c) indicate significant differences among groups at *P* < 0.05.

**Figure 2 microorganisms-10-02181-f002:**
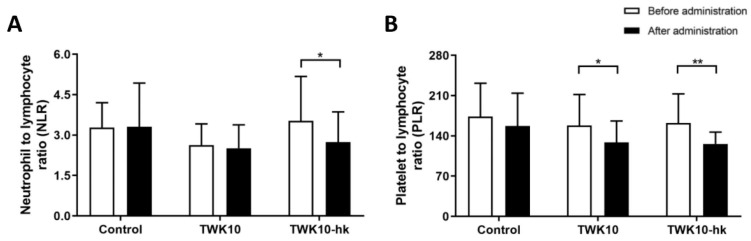
Effects of TWK10 on pro-inflammatory indicators after exercise challenge. Blood samples were collected and analyzed at 120 min after fixed intensity and period exercise challenges. (**A**) Neutrophil to lymphocyte ratio (NLR), and (**B**) platelet to lymphocyte ratio (PLR) were examined. Data are presented as mean ± SD. The significance of differences among groups were analyzed by two-way repeated-measures ANOVA with Tukey *post-hoc* test. Differences before and after administration were analyzed by two-way repeated-measures ANOVA with Bonferroni *post-hoc* test. * *P* < 0.05, ** *P* < 0.01.

**Figure 3 microorganisms-10-02181-f003:**
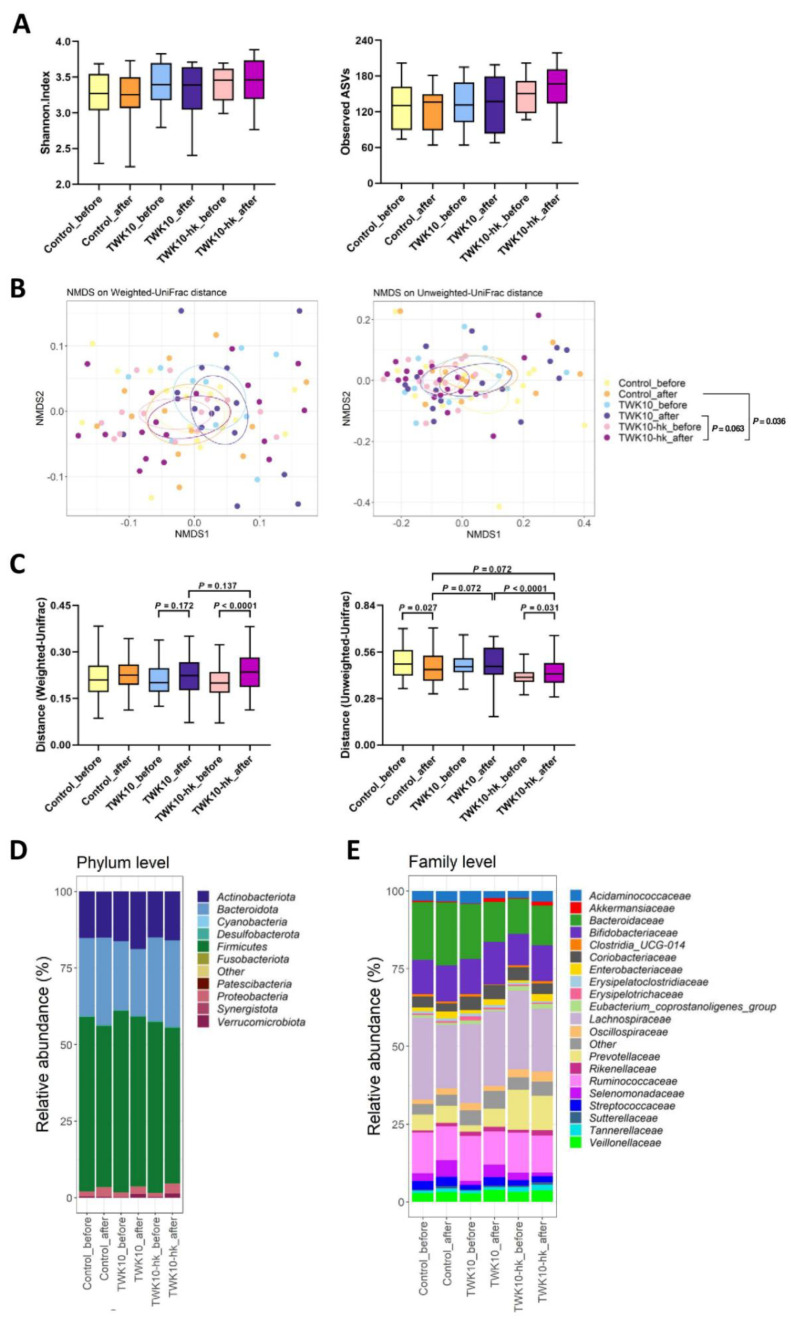
Comparisons of bacterial diversity among groups and its changes in response to intervention: (**A**) Box plots showing differences among the three groups (Control, TWK10, and TWK10-hk) in α-diversity indices (Shannon index and observed ASVs) before and after administration. Each box plot illustrates the median, interquartile range, minimum, and maximum values. ASV: Amplicon Sequence Variants; (**B**) NMDS plots of bacterial β-diversity based on weighted UniFrac distance (left panel) and unweighted UniFrac distance (right panel); (**C**) Distances of bacterial β-diversity based on the weighted UniFrac distance (left panel) and unweighted UniFrac distance (right panel). Comparisons of bacterial composition based on the top 10 phyla (**D**) and top 20 families (**E**) of bacteria in all samples. Others, remaining phyla or families with lower relative abundance.

**Figure 4 microorganisms-10-02181-f004:**
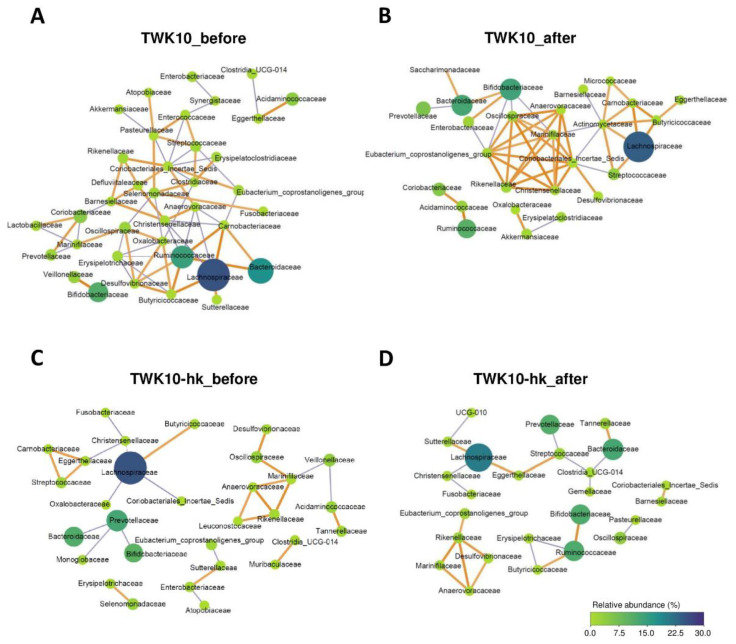
Co-occurrence network analysis of the gut microbiota. Bacterial networks were generated using SparCC correlation coefficients, based on relative abundances at the family level: (**A**) In the TWK10 group before administration (TWK10_before), there were 37 nodes, and correlation coefficients ranged from |0.60| to |0.92|; (**B**) In the TWK10 group after administration (TWK10_after), there were 28 nodes, and correlation coefficients ranged from |0.60| to |0.74|; (**C**) In the TWK10-hk group before administration (TWK10-hk_before), there were 30 nodes, and the correlation coefficients ranged from |0.61| to |1.00|; (**D**) In the TWK10-hk group after administration (TWK10-hk_after), there were 25 nodes, and correlation coefficients range from |0.60| to |0.85|. Nodes represent bacteria families; grey and orange color edges represent negative and positive correlation coefficients, respectively. The size and the degree of the green color of nodes in the network represent the relative abundance of each taxon in each group.

**Figure 5 microorganisms-10-02181-f005:**
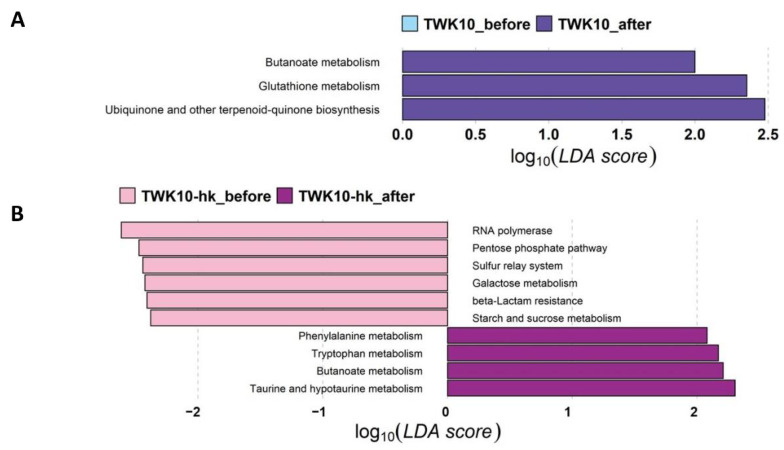
LEfSe analysis of metabolic function profiles using PICRUSt in subjects receiving viable or heat-killed TWK10. Linear discriminant analysis (LDA) effect size (LEfSe) analysis revealed significant differences in functional profiles between before (negative score) and after (positive score) administration in the (**A**) TWK10 and (**B**) TWK10-hk groups. LDA scores (log_10_) > 2.0 and *P* < 0.1 are shown.

**Figure 6 microorganisms-10-02181-f006:**
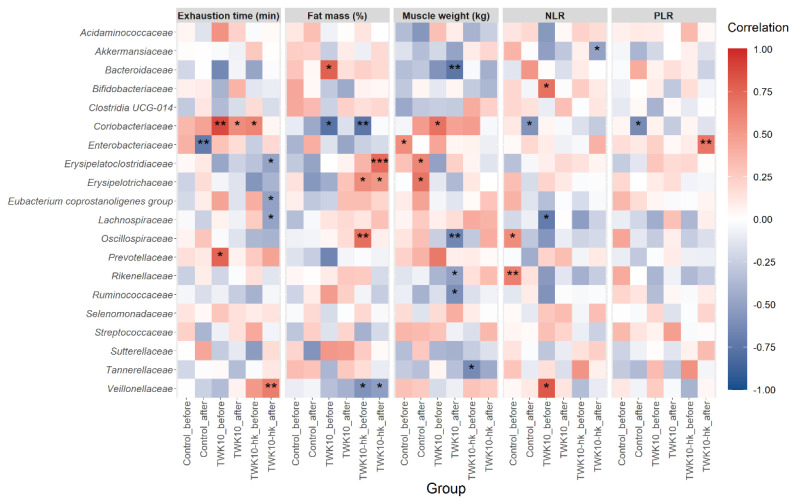
Heatmap of Spearman’s correlation analysis between gut microbiota and functional parameters in subjects who received viable or heat-killed TWK10 administration. Spearman’s correlation values were calculated between the abundances of the top 20 most abundant bacterial families and phenotypic changes following administration of viable or heat-killed TWK10. Red squares, positive correlations; blue squares, negative correlations. * *P* < 0.05, ** *P* < 0.01, *** *P* < 0.001.

**Table 1 microorganisms-10-02181-t001:** Characteristics of study subjects.

	Group		
	Control	TWK10	TWK10-hk
**Male**	*n* = 9	*n* = 8	*n* = 9
Age (y)	22.4 ± 1.9	22.0 ± 2.3	22.6 ± 3.5
BMI (kg/m^2^)	24.0 ± 2.4	24.5 ± 3.9	22.7 ± 3.8
VO_2max_ (mL/kg/min)	49.7 ± 7.5	49.0 ± 8.1	49.9 ± 9.8
Energy intake (kcal/day)	2686 ± 199	2649 ± 435	2557 ± 463
**Female**	*n* = 9	*n* = 9	*n* = 9
Age (y)	20.8 ± 1.1	20.7 ± 0.7	20.6 ± 0.7
BMI (kg/m^2^)	20.1 ± 1.7	22.8 ± 5.4	21.2 ± 3.5
VO_2max_ (mL/kg/min)	44.9 ± 9.1	44.8 ± 10.3	45.0 ± 10.6
Energy intake (kcal/day)	1831 ± 254	1855 ± 180	1760 ± 311

Data are presented as mean ± SD. Statistical differences among groups were analyzed by one-way ANOVA with Tukey *post-hoc* test.

**Table 2 microorganisms-10-02181-t002:** Effects of TWK10 on muscle weight and fat mass.

	Group		
	Control	TWK10	TWK10-hk
**Muscle weight (kg)**			
Before administration	27.46 ± 8.56	26.94 ± 6.69	25.12 ± 5.79
After administration	27.31 ± 8.25	27.62 ± 6.67 ***	25.26 ± 5.66
Change	–0.15 ± 0.67 ^a^	0.64 ± 0.66 ^b^	0.14 ± 0.51 ^ab^
**Fat mass (%)**			
Before administration	20.89 ± 8.57	24.84 ± 8.46	22.56 ± 8.67
After administration	20.80 ± 8.38	23.30 ± 8.07 ***	22.08 ± 8.74
Change	–0.09 ± 1.04 ^a^	–1.46 ± 1.52 ^b^	–0.48 ± 0.54 ^ac^
**BMI (kg/m^2^)**			
Before administration	22.06 ± 2.80	23.61 ± 4.69	21.97 ± 3.62
After administration	21.93 ± 2.66	23.36 ± 4.70	21.94 ± 3.49
Change	–0.12 ± 0.29	–0.23 ± 0.93	–0.03 ± 0.34

The changes in muscle weight, body fat, and BMI of the indicated groups were calculated as differences before and after administration. Data are presented as mean ± SD. Statistical differences among groups were analyzed by two-way repeated-measures ANOVA with Tukey *post-hoc* test. Statistical differences between values before and after administration were analyzed by two-way repeated-measures ANOVA with Bonferroni *post-hoc* test. *** *P* < 0.001. Significant differences in the changes between before and after administration in each subject among the three groups were analyzed by Kruskal–Wallis test with Dunn *post-hoc* test. Different letters (a, b, c) indicate significant differences among groups (*p* < 0.05).

**Table 3 microorganisms-10-02181-t003:** Effects of TWK10 on SCFAs in feces.

	Group		
	Control	TWK10	TWK10-hk
**Acetate (mM)**			
Before administration	7.17 ± 2.53	5.30 ± 1.96	6.48 ± 2.14
After administration	9.67 ± 5.25	8.56 ± 3.37 *	10.63 ± 6.17 *
**Propionate (mM)**			
Before administration	2.20 ± 0.44	2.00 ± 1.27	2.12 ± 0.88
After administration	2.67 ± 1.11	2.51 ± 2.16	3.80 ± 3.81 ^§^
**Butyrate (mM)**			
Before administration	0.96 ± 0.69	0.76 ± 0.56	1.11 ± 0.71
After administration	1.45 ± 0.86	1.15 ± 0.64 ^#^	1.33 ± 0.77

Data are presented as mean ± SD. The significance of differences among groups were analyzed by Kruskal–Wallis test with Dunn’s *post-hoc* test. Differences between values before and after administration were analyzed by Mann–Whitney *U* test. * *P* < 0.05, ^§^ *P* = 0.0857, ^#^ *P* = 0.0744.

## Data Availability

The datasets used and/or analyzed in the current study are available from the corresponding author on reasonable request.

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
