# Peer review of "Different Impacts of Heat-Killed and Viable Lactiplantibacillus plantarum TWK10 on Exercise Performance, Fatigue, Body Composition, and Gut Microbiota in Humans"

_microorganisms, 2022, doi:10.3390/microorganisms10112181_

Round 1
Reviewer 1 Report
I read this report on 'postbiotics' with interest.
Specific comments:
1. In the study title, it should be indicated that this was a pilot trial.
2. In addition to ref [2,3], suggest to also cite a review showing the benefits of probiotics on psychiatric disorders (citation: pubmed.ncbi.nlm.nih.gov/29197739).
3. How was the sample size determined? There was no evidence of power calculations. Was this meant to be a pilot trial given the small sample sizes.
4. This study should have been reported in accordance with established CONSORT guidelines. As per the guidelines, the trial should be registered and the registry number presented; a full trial protocol should be available online (indicate web address).
5. Please include a CONSORT flow diagram to show the enrolment, intervention allocation, follow-up processes, etc.
6. "All subjects were requested to maintain their usual diet and lifestyle" - were any of the subjects vegetarian or vegan?
7. The nature of replication in the experimental design is unclear, and the assessment of uncertainty in the reported measurement is absent or unclear.
8. In the discussion section, it is also important to highlight that the vast majority of research to date on the gut-brain axis has been conducted on animal models and much more human research is needed. It remains unclear whether similar events take place in humans too. Further mechanistic studies involving "omics" technologies, as adapted from previous studies (citation: pubmed.ncbi.nlm.nih.gov/30056340), might help shed light on these questions.
9. In the discussion section, authors should also mention that the gut microbiome is primarily studied using stool bacterial communities as a proxy. Stool samples are broadly representative of colonic luminal bacteria; however, by relying on stool samples, some communities of bacteria are neglected, including those from the small intestine and those embedded within intestinal mucosa (citation: pubmed.ncbi.nlm.nih.gov/34668228). Clear documentations of post-treatment events should be made mandatory, classified, and graded as in clinical trials.
10. The acknowledgments section was left blank.
Author Response
Dear Reviewer 1,
Thank you for your valuable comments on our manuscript.
We have completed the responses to your comments.
Please refer to the attachment. Thank you.
Best Regards,
chiachia

Reviewer 2 Report
In this paper the authors related the comparative effect observed before and after administration of a probiotic strain of Lactiplantibacillus plantarum TWK10, viable and heat killed in the regulation of exercises.
The manuscript is well written and is a very interesting work.
I only have little details, however, are only suggestions. I would like to congratulate the authors.
Is necessary to justify why authors choose the working concentration of alive and heat killed bacteria means 3 x 1011 UFC/day, and the time 6 weeks.
In the result section, the tittles need to represent the findings.
In the discussion section, it will be better if the authors withdraw the titles. And please withdraw the Fig referenced in this section too.
Line 554 to 557 Therefore, we speculate that the improvement of exercise performance mediated by heat-killed 555 TWK10 occurred through modulation of exercise-induced inflammation responses. Further investigations are still needed to understand the underlying mechanism.
In my opinion, with this results is very early to affirm this sentence, however authors end with the phrase …..Further investigations are still needed to understand the underlying mechanism. Mybe is better to compare with others works with heat killed bacteria and the microbioma study.
Line 626 to 628. These results indicate that probiotics supplementation can trigger microbial diversity in the gut and a consequent increased ability to produce SCFAs.
Is inminent compare the obtained result with references, there are many studies analyzing this field.
Line 644 to 646 Alterations in gut microbiota can stimulate differential 644 production of SCFAs (e.g., butyrate and acetate), which play important roles in maintenance of gut and metabolic health [89, 90]. The end of this paragraph is the same as the end of previous paragraph… It will be better if the authors joined the information, for not to be redundant. I mean is the same with other words.
Author Response
Dear Reviewer 2,
Thank you for your valuable comments on our manuscript.
We have completed the responses to your comments.
Please refer to the attachment. Thank you.
Best Regards,
chiachia

Round 2
Reviewer 1 Report
Thank you for the revisions.